# Harnessing digital technology to improve agricultural productivity?

**Arjunan Subramanian** [iD] *

Adam Smith Business School, University of Glasgow, Glasgow, United Kingdom

* arjunan.subramanian@glasgow.ac.uk

## Abstract

Can improving access to mobile extension improve agricultural productivity? Recent evidence suggests both significant and insignificant ways in which SMS-based agricultural information could affect farming outcomes. It is unclear if variations in the programs' design or the methodological challenges in evaluating the programs cause wide-ranging impacts. Extension hotline services provide rapid, unambiguous information by agricultural experts over the phone, tailored to time- and crop-specific shocks. Using methods from experimental economics, we randomly distributed the hotline number to generate exogenous variation in the access to farming information. We conducted our study among 300 farmers in the South Indian state of Karnataka. Our results show that eliminating informational inefficiencies increases farmers' average yields for a high-stakes pigeon pea crop that faced adverse aggregate shock. The impact on the yield is through the adoption of cost-effective and improved farming practices. However, we do not observe any effect on the crops that were not affected by the shock. Our findings reveal that advisory recommendations customized to time- and crop-specific shocks are associated with a greater impact on agricultural productivity.

## Introduction

Agriculture is central to sustainable development and poverty reduction [1, 2]. The context of agriculture worldwide has changed dramatically, and despite universal implementation and expansion of the agricultural extension program, significant challenges remain [3, 4]. In recent years, phone-based agricultural advisory services have evolved to take advantage of mobile phone networks' emergence and broader coverage [5–8]. Some studies have suggested both significant and insignificant ways SMS-based agricultural information could affect farming outcomes [8–10]. It is unclear if variations in the programs' design or the methodological challenges in evaluating the programs cause wide-ranging impacts [11].

One key concern for the mixed evidence is the selection bias that arises from the failure to control spillover effects and contaminate the control group. With access to SMS-based information on mobile phones, farmers can contact members of their social networks more efficiently, thereby intensifying the probability of spillovers among treatment groups. Furthermore, recent studies randomize the treatment at the household level, resulting in

**Data Availability Statement:** All relevant data are within the paper and its Supporting Information files.

**Funding:** The author thankfully acknowledges the support of the ESRC-DFID funded research project with ESRC Grant Reference: ES/J009334/1. The

views expressed in this paper are entirely those of the author and in no way represent either the official policy of funders or the policy of any other part of the UK government. The funders had no role in study design, data collection and analysis, decision to publish, or preparation of the manuscript.

significant information sharing between treatment farmers and their peers [9]. Information sharing between treatment groups can bias the effect of the intervention. Positive impacts are more likely attributable to the selection of farmers that are already better off, more efficient, better educated, and well informed than ascribable to the intervention's effect.

To examine the causal impact of phone-based agricultural advisory services on farming practices, crop production, and farm profits, we designed an intervention that minimizes the information spillovers by drawing samples from a broader spread of villages in the Indian state of Karnataka using clustered random sampling. The information treatment is randomized at a level (village) where subjects are less likely to interact, compete for workers, or share resources across treatment groups. The average distance between the closest control and treatment household within the same agro-climatic zone is 35 km.

To generate an exogenous farm-level variation of farmer hotlines, we randomly assigned an extension hotline number to a subset of farmers. The strategic objective of the extension hotline is to provide rapid, unambiguous information by agricultural experts over the phone to improve current farming practices. More than just access to information, the timing of the delivery of expert advice may be crucial in farming.

We study the impact of rapid agricultural information disseminated via telephone hotline on the productivity of four main crops–pigeon pea, finger millet, horsegram, and paddy–widely grown in central and southern India. The experimental region suffered from a natural disaster in the form of a devastating pest outbreak and disease at the end of the agricultural season in 2013. The pigeon pea crop in the study area was infected with sterility mosaic disease (SMD), which is one of the most devastating diseases of pigeon pea in India. The SMD is overwhelming and tends to affect the farming population collectively (covariate shock). Early damage control intervention, however, can salvage the crop loss from an SMD attack. Here, rapid and timely access to extension information can be enormously helpful.

Our main result shows that access to better farming information on damage control increases crop yield by 31% by adopting cost-effective and improved farming practices. Additionally, we have two key findings from the disaggregate analysis. First, access to the hotline increases pigeon pea yield by 87% for the treatment farmers relative to the control group with no access to the hotline. Second, though the overall costs of cultivation increased by 39% from intervention, profits for the treatment farmers cultivating pigeon pea are higher by 70%. The observed impact is understandable given the nature of pest shock in the pigeon pea crop and the timing and significance of the expert advice received via the hotline in reducing the losses for the treatment group.

Our study contributes to the evolving literature on the impact of digital advisory services in agriculture in two ways. First, we present new evidence on improving the delivery of agricultural extension services in developing countries [12–14]. In existing experimental studies, the provision of agricultural advisory services is still heavily supply-driven [13]. However, observational studies have shown that the demand for extension services may not be high [15, 16]. For instance, a recent study on Malawi suggests that access to agricultural advice does not necessarily lead to greater crop productivity [17]. Farmer perceptions of the usefulness and relevance of received advice are significant positive correlates of agricultural productivity outcomes. Thus, with low demand, the supply-side intervention with even good quality extension services is unlikely to influence crop yield, though also dependent on the type of crop and nature of crop shock. Our intervention provides advisory recommendations, delivered via a hotline, which is tailored to time- and crop-specific shocks to gauge the impact on agricultural productivity. Our results show that the demand for extension services increases when extension messaging relates directly to specific shocks and is provided during the window of time in which responses to those shocks are feasible.

Second, we provide evidence that failure to account for heterogeneity in crop-specific shocks may be a limiting factor in adopting improved technology. There is no empirical study to the best of our knowledge that assesses whether information accounting for heterogeneity in crop-specific shocks can result in higher yields and profits. This paper presents evidence indicating that customized shock-specific recommendations could improve crop productivity relative to existing generic recommendations [18, 19]. Our result is consistent with recent evaluations suggesting that personalized, localized advice is far more effective than blanket recommendations [13, 14]. We differ from these studies that focus on digital advisory services provided via extension agents in the field, while our evaluation utilizes hotline phone services.

## KCC hotline services

India's agricultural extension system is one of the most extensive public sector knowledge and information dissemination institutions globally. The success of this system during the Green Revolution is well documented [20]. However, over time, the public extension system has evolved into a nodal organization to distribute subsidized farm inputs under various agricultural development programs. Consequently, due to this shift in focus, government extension services' significance and effectiveness have fallen drastically. Specifically, in Karnataka, only 11.5% of the farming households had at least one contact with a government extension worker in the 2003 survey year [21]. Our baseline survey also reports similar access to agricultural extension services. In the agricultural year, 2012–13, only 8% and 4% of the farming households had one or more visits from extension workers and scientists.

Despite reforms to strengthen extension and research systems, several performance issues still hinder the effectiveness and efficiency of the public agricultural extension system in India [22]. Farmer hotline services were introduced as part of the new agricultural development programs to take advantage of India's broader mobile phone network coverage. For instance, in early 2004, as part of a policy to rapidly deliver agricultural extension services to farming communities across the country, the Department of Agriculture & Cooperation (Ministry of Agriculture) introduced the *Kisan* Call Centre (KCC) hotline services. Their purpose is to respond promptly to farm-related problems, improve the quality, and accelerate the transfer and exchange of information to farmers.

Under the KCC hotline services, farmers call a typical, toll-free advisory phone number to access expert advice from Level 1 operators (agricultural graduates) in 13 regional centres. The crop-specific extension information, provided from centres located across the country, is delivered in 21 local languages. Further, queries from Level 1 operators are supported by Level 2 experts, who are in different parts of the country at State Agricultural Universities, Indian Council for Agricultural Research Institutes, and State Department of Agriculture [23]. Farmers speak live with the Level 1 operators and get answers to their questions immediately. If Level 1 operators cannot provide answers, the calls are passed on to Level 2 experts who promptly offer answers. The solutions provided are plot- and crop-specific.

Despite several years of operations by KCC, there is surprisingly no rigorous impact evaluation of its services. This paper provides the first thorough evaluation of the KCC hotline in Karnataka. Traditional public sector extension services are known to be ineffective, and KCC initiatives have not yet widely spread. Our baseline survey reveals that farmers in this region were unaware of the KCC hotline number. We exploit the inadequate coverage of the KCC hotline in this region to design a field experiment to evaluate its effectiveness.

## Nature of crop pest shock

To study the heterogeneity in the impact of hotline across crops, we examine the differences in the type of shock affecting the crops. At the end of the agricultural season (*Rabi* 2013), SMD infected the pigeon pea crop in the study area. The *Kharif* season is from June to September, and *Rabi* is from October to January. Pigeon pea is usually sown in June or July, but it could be sown up to the end of August under delayed monsoon rain conditions. The cropping duration of pigeon pea could range from six to nine months and sometimes even longer depending on the seed variety, whether it is early or long-duration maturing. The disease is contagious and can spread rapidly across cultivated areas. In other words, the SMD that attacked the pigeon pea crop in the experimental area is devastating and tends to affect all farms in the vicinity and beyond [24, 25]. The survey results reported in S2 Table shows a balance of the covariate shock across treatment groups indicating that SMD changed both the treated and control group equally.

However, the agricultural experts' advice could curtail the consequences of the disease [26, 27]. Though on average, farmers have 30 years of farming experience in the region of study (S4 Table), knowledge on the development of resistance by the pests and identification of appropriate chemical sprays to neutralize these pests pose a considerable challenge. Here, the agricultural experts' support is crucial, who have the relevant scientific knowledge of pest, and the chemical composition of currently available sprays in the market. Unfortunately, despite knowing that local fertilizer and pesticide shops have perverse incentives, farmers rely too much on these shops for information on sprays, who most often recommend ineffective and expensive sprays.

Addressing the disease requires better coordination between farmers on the application of sprays and other control measures. Careful roguing of the infected plant is a restorative treatment measure to remove the source of infection cost-effectively. The yield loss, however, depends on the growth stage at which the infection occurs. When the plant is less than 45 days old, the loss could be between 95%-100% in the early infection stage. In the case of late infection, after 45 days but early diagnosis, though dependent on the infection level, loss in yield can range from 26%-97%.

The infection of SMD in the study region, for most cases, became apparent after 90 days of sowing but was diagnosed within one and two weeks after infestation. So, there seemed considerable scope for recovering crop yields, especially timely intervention with accurate diagnoses of disease and information on the application and dosage of relevant sprays and the cost-effective management of pest populations.

## Technology and management strategies disseminated

Though SMD is not a new disease, its causal agents have remained elusive to identification and characterization over many decades. SMD dynamics are influenced by many abiotic and biotic factors, diverse agricultural systems, and environmental conditions. Information on off-season survival of virus and mite vector, their spread during the cropping season from crop to crop, and within the crop and variation in disease incidence in a region is critical for understanding disease ecology. It is challenging considering that the crop is mainly grown in marginal farming systems with divergent cropping practices.

Several methods were recommended by the hotline to reduce SMD incidence by using pesticides to delay the onset of infection and disease spread, control through cropping management practices, and host-plant resistance. In the worst cases of the disease, the helpline recommended many chemicals to control the mite vector and minimize the disease's spread (S5 Table). Correct timing and dosage are critical for effective control of the vector populations.

The helpline relayed to farmers not to go for continuous pigeon pea cultivation in the same field and treat the seeds with Trichoderma Viride at 4 g per kg of seed. Farmers were advised to pull out the infected plants and burn them as SMD spreads quickly—roguing or pruning diseased plants were instructed to promote cost-effective treatment. Neem oil spray (1 to 2 ml per litre) or Profenophos (2 ml per litre) was recommended instead of expensive, strong chemicals. Farmers were advised not to spray chemicals inadvertently but adopt methods like pruning the infected plants and only react with chemicals when the pest population reached the optimal threshold. The threshold was described, shown, and discussed with farmers.

Overall, removing diseased plants from the field (roguing) was the highly recommended and widely adopted SMD management measure. Roguing is most effective, easier to perform, and has a less detrimental effect on productivity when carried out in early crop infections. The recommendation is likely to reduce the cost of production relative to the control group. More specifically, it will reduce the cost of insecticides and associated labour use. In contrast, the information treatment recommendations are likely to raise the cost of intercultural operations with the increased labour use for regular inspection of the crop and roguing of infected plants. The damage control recommendations from the advisory services can result in greater yield and increase the harvesting cost of labour relative to the control group.

Some pigeon pea varieties offer broad-based resistance to SMD, while others are highly susceptible to the disease. For types that were highly sensitive heavy doses of chemicals were prescribed whilst roguing for resistant varieties. Thus, the information provided by the hotline varied depending on the crop variety.

Non-pigeon pea crops suffered from many isolated incidences of minor pest problems like shoot flies, stem borers, and leaf-sucking insects. The IPM methods over the hotline can be equally effective against these minor pests and diseases, but farmers chose to access the hotline primarily to address SMD in pigeon pea. The low demand for hotline by farmers who cultivated only non-pigeon pea suggests that farmers value the extension information more when faced with crop loss risk (S3 Table).

## Materials and methods

### Sample selection

We conducted our study among farmers in Gubbi *taluk* (sub-district), located in the Tumkur district, about 70 kilometres from Bangalore, in the south Indian state of Karnataka. The ethics committee at the Indian Institute of Management in Bangalore, India, approved the study. A written letter of consent was obtained from the ethics committee. Written consent was also obtained from all the study participants. While Tumkur was one of the districts preselected for implementing a large-scale agricultural program (S1 Description in S6 Table), we chose Gubbi for two reasons. (a) After completing the baseline survey under the agricultural program, we noticed the early signs of an SMD outbreak in Gubbi. (b) Access to the KCC hotline is available, but farmers are unaware of its services. The Gubbi sub-district is representative of pigeon pea growing areas across central and southern India. Any generalizations of the results beyond this region may be problematic. A study from the region suggests that only 87% of households own mobile phones [28]. However, our baseline survey indicates 100% ownership among sample households owning one or more mobile phones (See S4 Table).

To randomize farmers into treatment, we followed a three-stage procedure. In the first stage, we stratified the 327 villages using pre-experimental data on the type of crops predominantly grown. We used crop area and main staples grown by the households to identify the crops. Two crops–pigeon pea and finger millet–proved significant in terms of area and main staples. To guarantee the desired heterogeneity in terms of crops, we stratified villages by their

share of each crop area to total cultivated into three predominantly pigeon pea-growing, finger millet-growing, and the rest. Twenty-eight villages, primarily growing pigeon pea, were randomly allocated to treatment, with twenty-five to control. We randomly assigned twenty villages with a significant share of finger millet to treatment and twenty-two villages to control. The rest of the twenty-six villages are split, with eleven randomly allocated to treatment and the rest to control. The experimental households in each stratum consist of 90 pigeon pea farmers, 107 finger millet farmers, and 38 growing other crops.

In the second stage, we randomly allocate 59 villages of the 327 villages in Gubbi *taluk* to treatment and 62 villages to the control group (Fig 1). In the third stage, we randomly selected households within the selected villages. Due to extenuating circumstances, households were purposely oversampled by randomly selecting pigeon pea growing farmers at the village level. As previously stated, the average distance between the closest control and treatment household is 35 km. This procedure minimizes the flow of information to the control group.

From the Bhoomi database, a census of land ownership in Karnataka, we randomly sampled 300 households from 121 villages with 200 treatment and 100 control households. Bhoomi database is an outcome of the Bhoomi project for the online delivery and management of land records in Karnataka. The state government of Karnataka implemented the project to digitize all the manual Record of Rights, Tenancy and Crops (RTC). Our power calculations based on the baseline survey in the DATES program (S1 Description in S6 Table) guides the sample size we use in this study. We identified a total area of 499 acres belonging to 300 farmers, who cultivated four main staple crops widely grown by farmers in this region for both market and home consumption. Selected farming households produced more than one type of

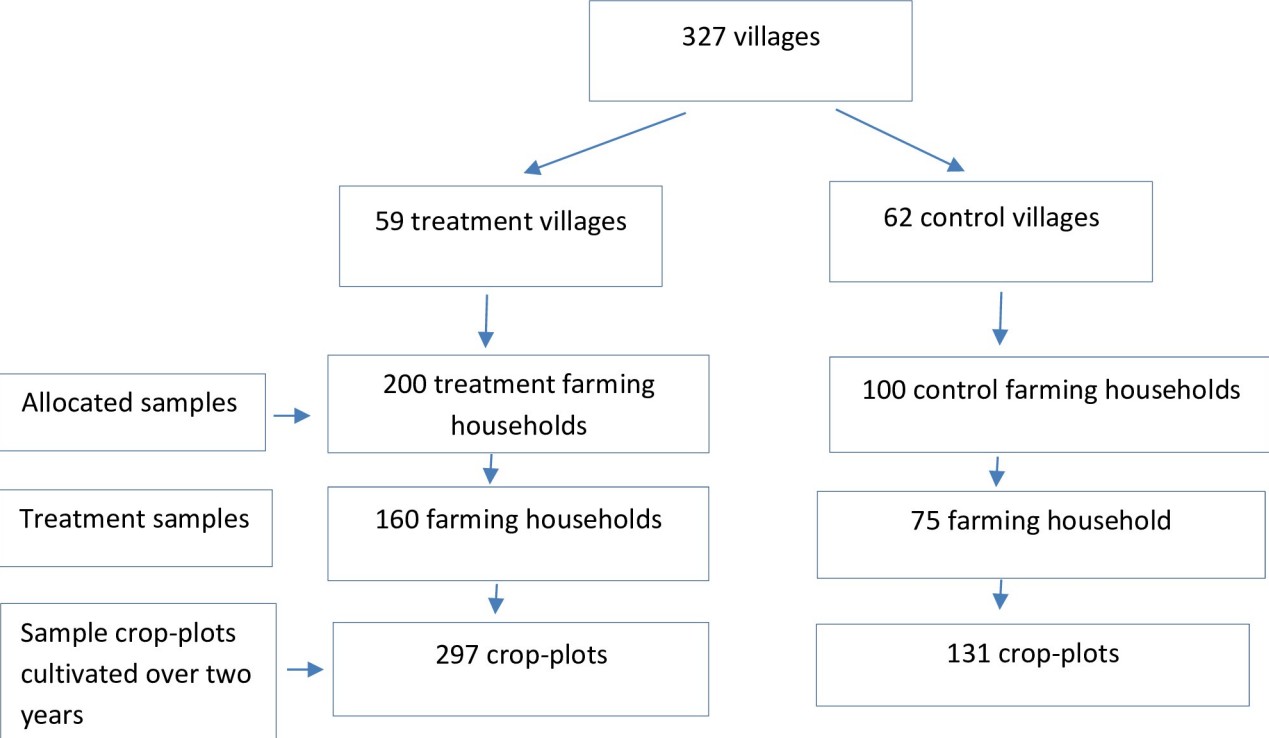

**Fig 1. Experimental design.** Notes: Treatment samples refer to farming households who were provided with phone numbers to access support for extension information. Due to extenuating circumstances, households were purposely oversampled by randomly selecting pigeon pea growing households at the village level.

crop from two crop cycles between June 2012 and June 2014, which could be up to four differ-
ent crops or the same crop but of a diverse variety of seeds (S1 Fig).

## Data

The dataset we work with is an unbalanced panel of crop-plots of varying plot size over two
years. Here crop-plot refers to a parcel of land with a single crop demarcated by raised bunds.
Though the samples selected are at the household-level, the data collected is at the crop-plot
level. Hence, our estimation strategy is not at the household level but disaggregated by crop-
plots. In the baseline survey, there are 193 crop-plots which also appears in the endline survey,
along with 42 additional crop-plots. Fig 1 shows that 160 treatment farmers cultivate 297 crop-
plots, and 75 control farmers cultivate 131 crop-plots over two years. The breakdown of crop-
plots by type of crops are 148 pigeon pea, 208 finger millet, and 72 other crops.

In June 2013, the trained enumerators visited the sampled households at home and farm to
administer a baseline survey. Our baseline survey collected retrospectively detailed crop culti-
vation information, such as outputs and inputs used, for all the crops grown during the 2012–
2013 crop cycle (namely, crop cycle 1 in S1 Fig). Other information collected includes farmer
specific characteristics, household structures, and details of any extension information
received and their sources.

While carrying out the baseline survey starting in June 2013, we distributed the KCC hot-
line number to the treatment farmers. After completing each household survey, the enumera-
tors handed the KCC number to the household, talking through its significance and how to
use it. The hotline provided treated farmers with access to real-time farming information to
adopt cost-effective and improved farming practices. After distributing the KCC number, we
could not reach two households, so we had to drop these households leaving a sample of 198
treatment households. Immediately after the harvest, we conducted the endline survey
between June and July 2014. In the S1 Description in S6 Table, we provide a detailed descrip-
tion of the experimental intervention.

While some crops may be inherently non-responsive to extension information, others may
be conducive to recent scientific development in agronomic practices. The nature of the exten-
sion services is also evolving from extensive use of fertilizer and pesticides during the Green
Revolution period to the prudent use of harmful inputs and better management of the pest.

In the endline survey, we administered a detailed questionnaire asking for information on
KCC number usage. The questions include how many calls were made to seek which informa-
tion, how effective the response was, the nature of problems faced–whether suffered from pest
attack and weed infestation, whether rainfall was below normal and the expected yield loss.
We asked additional questions and modules from the farm and household surveys adminis-
tered in the baseline survey. The farm survey includes a production module that records the
crop outputs for the months preceding the survey interview. We collect the type of crop pro-
duced, the area planted, output quantity and prices, and the duration of the crop produced.

We collect total labour hours worked (family and hired) in the cost module, input quantity
and prices, and revenues. We record the information for each crop and farming operations.
We calculate profit as the total revenue net of the total cost, considering all the input costs,
including family labour (quantified by going market wage in the local labour market). The
household survey module records the farm households' demographic characteristics such as
member-wise information on age, sex, education, occupation, salary and wage incomes earned
from agricultural and non-agricultural employment, and details of assets owned.

The usage of the KCC helpline number was high, with 81% of the treated households calling
the number to receive a range of information. Note that the attrition among our samples is

low, with only two households dropped out. Among control, only three farmers reported having accessed the helpline number despite not receiving the number. In S3 Table, we report the mean of the number of calls made to the helpline number and the self-reported effectiveness of the response by information type. We asked farmers to keep a record of all the calls they make and the information they received. Many farmers reported that they had to call several times because their calls got disconnected. If the problem remained unresolved in the first stage, they had to wait longer for the Level 2 expert's advice. Sometimes the wait could be even for 30 minutes, so the caller is asked to call back later.

The hotline provides solutions to farming-related problems faced by the farmers for all crops. However, most calls made were concerning pigeon pea to obtain information on pest and diseases (42 on average across treatment farmers) and to ask for appropriate cultivating practices to contain them (45 on average across treatment farmers). Since the treatment farmers received the hotline number only after crop cycle 1 (See S1 Fig), the information received can be related only to crop cycle 2. Moreover, the timing of the information farmers obtained from the helpline–that was about the agricultural operations for the standing crops–coincides with the SMD occurrence. The call timings from the call records also indicate that the farmers' requested information was about SMD.

The farmers who received the information felt the advice was helpful in most cases: 92% of the farmers who received information on pest and diseases report the advice was helpful. In S5 Table, we summarise the details on the type of information requested, reasons for the request, and advice received from the helpline to deal with the problem. Overall, the summary shows that farmers asked for a range of information from land preparation to chemical sprays, and across different crops, including plantation crops.

## Regression model

Our randomization procedure allows us to use a straightforward estimation strategy to assess the impact of information provision via the hotline on production outcomes:

$$Y_{ijt} = \beta_0 + \beta_1 A_j + \beta_2 T_t + \beta_3 A_j T_t + X'B + \varepsilon_{ijt} \tag{1}$$

where $Y_{ijt}$ is the outcome of interest in crop $i$ of plot $j$ in period $t$; $A_j$ is a dichotomous variable, equal to 1 if plot $j$ belongs to the treatment group in both crop cycles (farmer owning the plot would receive the KCC hotline number); $T_t$ is a dichotomous variable that takes value 1 for crop cycle 2; $X$ is a vector of control variables, and $\varepsilon_{ijt}$ is an error term. Coefficients $\beta_1$ is the strata fixed effect. Here $T_t$ is included to distinguish the two crop cycles over which the yields are compared. Note that $\beta_2$ is the time fixed effect. Additionally, we also include three interaction terms for each crop to examine each crop's yield impact. Further, we add separate crop-wise dummies interacted with hotline impact ($\beta_3$) to examine the crop-specific effect. Next, we present results from the disaggregate analysis for the two dominant crops.

The effect of interest ($\beta_3$), capture the treatment's average impact relative to the control group (intent-to-treat, ITT). Note that our analysis uses the initial treatment assignment and not the actual information treatment. Since we compare the yields of crops from two distinct crop-cycles, we use a difference-in-difference estimation strategy to study the impact of hotline treatment. The idea underlying this estimation strategy is to compare the outcome before and after the SMD shock and between treatment and control groups. The difference-in-difference approach accounts for time-invariant unobserved heterogeneity and non-zero adoption status at baseline [29].

The standard errors are clustered at the village level in all individual plot level regressions. Drawing inference employing clustered standard errors with a low number of clusters can be

unreliable; we apply a wild bootstrap to bootstrap the T-statistics with each cluster [30]. We examine crop yields' recovery using crop-wise individual farm plot level, unbalanced panel data gathered from detailed farm surveys. As the intervention was at the crop-plot level within each household, we analyze it at the crop-plot level. We compare the same crop-type over the two crop-cycles grown in the same plot since we do not observe crop-rotation.

## Results

### Descriptive statistics and randomization check

Baseline balance checks between the treatment and control groups, presented in the S4 Table, show overall balance except for one variable. Most farmers report normal rainfall in the baseline (first crop cycle). Both disease and weed infestation was low, resulting in fewer sprays across the treatment groups. The differential effect of shocks to output also shows balance across the treatment and control groups.

Note that we compare two different crop cycles to determine the effectiveness of the helpline. Although our results suggest no significant differences between the treated and control households at the baseline, we also need to be assured of the balance of different shocks to output in the second crop cycle. For instance, the pest and disease, rainfall, and weed infestation did not affect the control and treatment groups differentially.

In S2 Table, we report the differential effect of three shocks to the crop–weather, pest/disease, and weed infestation. Most farmers report normal rainfall in the second crop cycle, which was also the case with the first crop cycle. Compared to the first crop cycle, the second crop cycle had much higher pest and disease incidence, while weed infestations were the same. We observe a high prevalence of the disease in the pigeon pea affected by the sterility mosaic disease (SMD). Overall, results show the balance across treatment groups for the different shocks to output.

### Effect of KCC hotline phone

The intent-to-treat impact on the hotline in the regression with crop yield as the dependent variable is significantly positive (Table 1, column 1). If hotlines help farmers with better farming information–mainly on adopting cost-effective and improved farming practices–we conjecture that hotlines can reduce crop yield losses. Our results show that hotlines are associated with a 31% increase in crop yields.

We next consider whether the effect of the hotline on crop yield differs by crop types. Because different crops may be affected differently by shocks and farming practices, heterogeneity in the hotline's impact may conceal the overall effects. We allow for the impact of the hotline to vary with the type of crop. As shown in column 2, we include three crop interaction variables, excluding paddy. The hotline has a substantial impact on pigeon pea management when both control and treatment farmers experienced pest shock. The effects of hotlines on finger millet and horsegram are not statistically significant. This finding suggests that a hotline can affect access to farming information differently by type of crop. Note the hotline's effect was strongest when the messages were indeed "hot"–when they were about urgent pest and disease management.

We now present the results from the disaggregate analysis by splitting the total sample into two dominant crops–pigeon pea and non-pigeon pea. Here, the idea is to compare the helpline's impact when a crop suffers from a covariate shock (pigeon pea) and crops that did not (finger millet, horsegram, and paddy; henceforth, non-pigeon pea). In column (3) for pigeon pea, we report a significant positive impact of hotline services on crop yield. Access to the hotline increased the treated farmers' yield compared to the control farmers, where both the groups are equally affected by the pest attack. Results for the non-pigeon pea, reported in

**Table 1. Regression of crop yield on hotline impact, crop-specific hotline impact, and baseline survey controls.**

| Dependent Variable: Log of Crop yield | Crop-plot level | | | |
|---|---|---|---|---|
| | Aggregate impact | | Disaggregate impact | |
| | All crops | Crop-wise | Pigeon pea | Non-pigeon pea |
| | (1) | (2) | (3) | (4) |
| Hotline impact | 0.313*** | 0.303* | 0.8732** | 0.0587 |
| | (0.097) | (0.093) | (0.3529) | (0.2531) |
| Hotline impact # pigeon pea | | 0.295*** | | |
| | | (0.055) | | |
| Hotline impact# finger millet | | 0.268 | | |
| | | (0.167) | | |
| Hotline impact # horsegram | | -0.172 | | |
| | | (0.207) | | |
| Constant | 1.434*** | 2.824*** | 0.3054 | 0.3403 |
| | (0.422) | (0.511) | (0.9737) | (0.3808) |
| Baseline survey controls | YES | YES | YES | YES |
| Year FE | YES | YES | YES | YES |
| Clustered SE | YES | YES | YES | YES |
| *R*-squared | 0.192 | 0.621 | 0.372 | 0.263 |
| Observations | 312 | 312 | 88 | 224 |

Notes: Hotline impact is the use of a hotline, conditional on treatment. Note that taking the log of crop yield reduces the number of observations from 428 to 312 because 116 crop-plots produced zero crop yields. Non-pigeon pea includes three crops: finger millet, horsegram, and paddy. Column (2) also includes three crop dummies, excluding paddy. Baseline survey controls include public/private sources of crop information; the number of visits of the public extension advisor (Ref: zero visits vs one visit, two visits); the number of years of crop farming experience (Ref: <16 years vs (16, 30 years), (30, 45 years) and > 45 years); the number of years of education (Ref: <6 years vs (6,10 years) and > 10 years); whether belonging to schedule caste/tribe; the log of total land owned in acres; log of total durable asset value; time (T) and treated (A). Wild bootstrap standard errors in brackets clustered by village code

* $p < 0.1$,

** $p < 0.05$,

*** $p < 0.01$.

columns 4, show no significant access to the hotline. This differential effect of the hotline is not unexpected because the pigeon pea suffered from covariate shock while the other crops had a regular agriculture season.

Since we observe productivity gains among pigeon pea farmers, we now dwell more in-depth into the intervention's impact on the pigeon pea crop across treatment groups. Fig 2 displays the estimated kernel density of yield by treatment status for the year 2014. A clear rightward shift in the distribution of pigeon pea yield is observed for the treatment farmers. Thus, there is a noticeable increase in pigeon pea yield for plots cultivated by farmers with access to farming information using the hotline.

The hotline advocating damage control resulted in 58% higher output per acre for the mean treatment farmer than the mean control farmer (Fig 3). Greater production would mean an increased cost of transplanting, weeding, threshing, and harvesting. Thus, the overall cost per acre increased by 39% for the mean treatment farmer relative to the mean untreated farmer. Despite the increase in the total cost, profit (which is the difference between output and cost) increased by Rs. 1752 per acre–or around 70% for the mean treatment farmer compared to the mean control farmer.

We now explore the potential mechanisms that resulted in the differential effect of the hotline. Is it likely that the pigeon pea is more responsive to the information provided by the

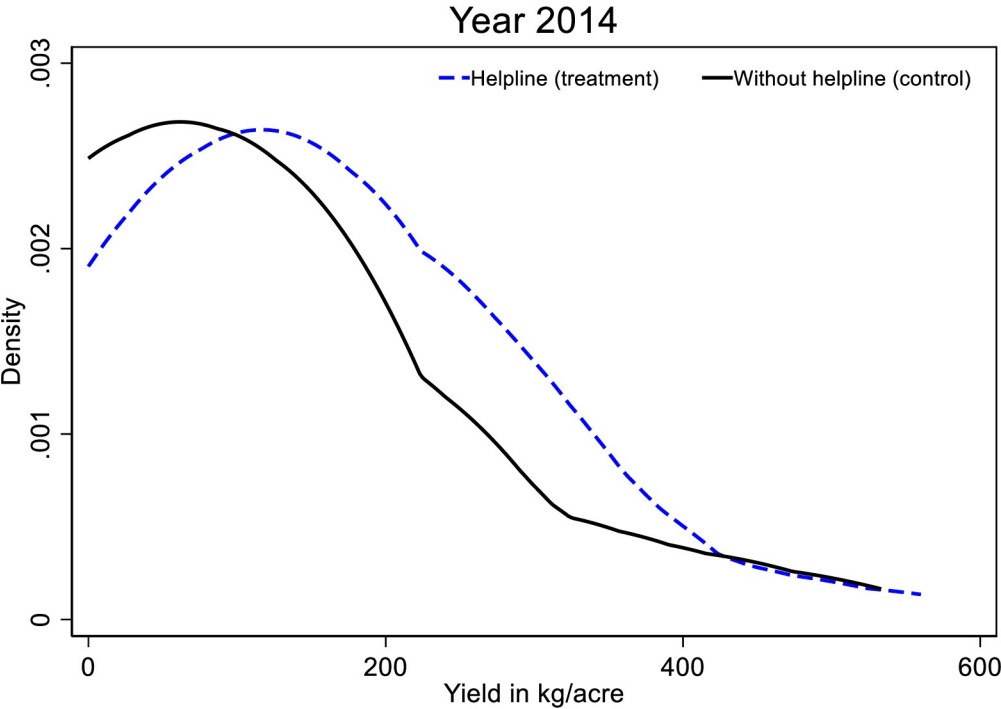

**Fig 2. Kernel density of plot-level pigeon pea yield by treatment status.** Notes: Density is estimated across all the pigeon pea plots. The dashed line is the density for all the pigeon pea plots cultivated by farmers who received the hotline number. Solid lines are density for all the pigeon pea plots grown by control farmers who did not receive the hotline phone number.

hotline? The hotline information–recommendations based on Integrated Pest Management (IPM) methods–is expected to be equally valuable for all crops and against most pests and diseases. Thus, the shock to pigeon pea *per se* is unlikely to cause differences in the crop yields. Since hotline information is crop and variety specific, the differences in variety (genetic diversity) should not matter for the outcomes.

Table 2, column (1), reports the hotline's effect on (log) total crop revenue by aggregating all four crops. The coefficient shows that the hotline treatment increased revenue by 49% relative to the control, which is significant at a 1% level. The total cost of cultivation, reported in the next column, shows that the hotline treatment also increased the cost by 39% for the treated relative to control. Note that total cultivation cost includes the cost of ploughing, harrowing, sowing, transplanting, interculture operations, weeding, micronutrients, insecticide, manure, fertilizer, threshing, and winnowing.

We next examine the drivers of the yield and revenue gains attributed to the treatment. We explicitly consider the cost of cultivation, splitting the total cost to investigate only three of the costs where hotline intervention is expected to make an impact. The analysis aggregating all the crops are estimated and reported in columns (3) to (5). We do not find a robust effect of the hotline on the cost of insecticide sprays and intercultural operations.

We next dis-aggregate individual crops' total costs to examine the changes in the cultivation practices likely attributable to the information from the hotline. Since the hotline's impact on finger millet and horsegram was not statistically significant, we did not report the results. However, it can be requested. Overall, the hotline seems to have increased pigeon pea's cultivation cost by 60% for the treated households relative to the control with no access to the hotline (column 6). The hotline advocating the targeted spraying of insecticides for the treated farmers

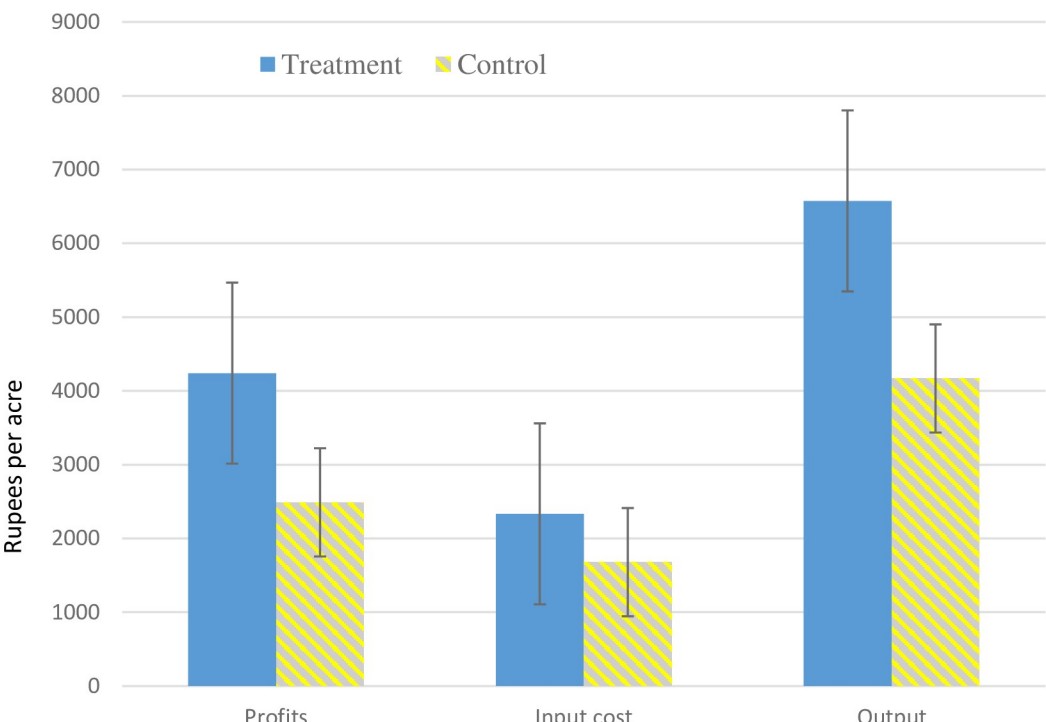

**Fig 3. Profits, input cost, and output for the mean pigeon pea farmer.** Notes: Input costs are in Indian rupees per acre that include the cost of ploughing, harrowing, sowing, transplanting, interculture operations, weeding, micronutrients, insecticide, manure, fertilizer, threshing, and winnowing. The input costs for the mean treatment farmer is 39% higher compared to the mean control farmer. The output per acre is 58% higher for the mean treatment farmer relative to the mean control farmer. Profits in Indian rupees per acre are the output net of input costs per acre. Profits for the mean treatment farmer was 70% higher compared to the mean control farmer.

reduced the cost of sprays by 88% relative to the control farmers (column 7). As expected, with no access to helpline information, the control farmers affected by the SMD typically reacted by over-spraying expensive chemicals.

However, the hotline increased the cost of intercultural operations by 86% for the treated farmers relative to the control farmers who had no access to the hotline. The IPM methods on damage control advocated by the extension hotline required treatment farmers to carefully rogue (prune) infected plants to remove the source of infection cost-effectively (See S5 Table, Column 1 SNo. 10). Since roguing (pruning) diseased plants is labour-intensive, it increased the cost of interculture operations. Consequently, the damage control, which resulted in a better harvest, also raised the overall cost of harvesting by 71% (Column 5).

Our analysis, so far, has examined the impact on each outcome individually for the many outcomes. The consideration of several outcomes raises questions about multiple hypothesis testing. In S6 Table, we examine which of our results are robust to different corrections for multiple outcome testing. We calculate the bootstrapped estimates of adjusted p-values [31]. Except for the log of production cost, we reject the null hypothesis that the coefficient is equal to zero at a 5% significance level.

## Discussion

Governments in the developing world are increasingly acknowledging the role of information and communication technology (ICT) in enhancing agricultural productivity and reducing poverty. Knowledge on harnessing the impact of ICT-based extension on agricultural

**Table 2. Regression of cultivation practices on hotline impact, pigeon pea hotline impact, and other baseline survey controls.**

| Dependent variable: | Crop-plot level | | | | | | | |
|---|---|---|---|---|---|---|---|---|
| | Log of revenue | Log of cultivation cost | Cultivation cost | | | Pigeon pea | | |
| | | | Log of insecticide cost | Log of interculture cost | Log of harvesting cost | Log of cultivation cost | Log of insecticide cost | Log of interculture cost |
| | (1) | (2) | (3) | (4) | (5) | (6) | (7) | (8) |
| Hotline impact | 0.488*** | 0.390** | 0.626 | 0.152 | 0.712** | | | |
| | (0.134) | (0.168) | (0.457) | (0.109) | (0.240) | | | |
| Hotline impact # pigeon pea | | | | | | 0.601** | -0.879** | 0.858*** |
| | | | | | | (0.317) | (0.364) | (0.321) |
| Constant | 8.437*** | 10.398*** | 6.308*** | 6.691*** | 7.827*** | 9.437*** | 4.483*** | 5.895*** |
| | (0.583) | (0.615) | (1.475) | (0.839) | (0.834) | (0.405) | (1.078) | (0.366) |
| Baseline survey controls | YES | YES | YES | YES | YES | YES | YES | YES |
| Year FE | YES | YES | YES | YES | YES | YES | YES | YES |
| Clustered SE | YES | YES | YES | YES | YES | YES | YES | YES |
| R-squared | 0.202 | 0.181 | 0.176 | 0.240 | 0.298 | 0.228 | 0.228 | 0.342 |
| Observations | 250 | 328 | 94 | 191 | 233 | 94 | 94 | 191 |

Notes: Baseline survey controls include public/private sources of crop information; number of visits of the public extension advisor (Ref: zero visits vs one visit, two visits); number of years of crop farming experience (Ref: <16 years vs (16, 30 years), (30, 45 years) and > 45 years); number of years of education (Ref: <6 years vs (6,10 years) and > 10 years); whether belonging to schedule caste/tribe; log of total land owned in acres; log of total durable asset value; time (T) and treated (A). Wild bootstrap standard errors in brackets clustered by village code

* $p < 0.1$,

** $p < 0.05$,

*** $p < 0.01$.

productivity is still limited. Our study empirically investigates the causal effect of phone-based extension on crop yields and profits. While our study confirms previous findings of a positive impact, our results distinctly show that this effect is only for a high-stakes crop that faced adverse aggregate shock.

More specifically, we show that access to better farming information on damage control increases yields by adopting cost-effective and improved farming practices. The treatment recommendations raised the cost of intercultural operations with the increased labour use for regular inspection of the crop and roguing of infected plants. The damage control suggestions from the advisory services resulted in greater yield and increased the harvesting cost of labour for the treatment group relative to the control group.

We have two key findings from the disaggregate analysis. First, results show that the hotline had no significant impact on finger millet, horsegram, and paddy yields. Similar results of no or moderate effect have previously reported in the literature from both experimental and non-experimental studies [4–11].

Second, access to the hotline increases pigeon pea yield for the treated farmers relative to the control group with no hotline access. Though the overall costs of cultivation increased from intervention, the pigeon pea treatment farmers' profits are higher. The nature of pest shock in the pigeon pea crop and the timing and significance of the expert advice received via the hotline reduced the losses for the treatment farmers. Note that we observe no significant statistical difference in pest attack incidence between treatment and control farmers. The timing of expert information delivery is highly crucial during farming distress that may vary from year to year.

On average, we find that farmers adopt cost-effective and improved farming practices only when their crop suffered from covariate shock. However, information on IPM methods from hotlines is equally helpful for all crops. The changes in the cultivating practices based on information from the hotline only for the pigeon pea crop suggests that farmers value the extension information more when faced with the risk of crop loss. This study documents that ICT tools do not unequivocally increase productivity, so merely scaling up the ICT infrastructure for improving access to extension information may be insufficient to enhance agricultural productivity. For augmenting the demand for extension services, hotlines can deliver locally customized information at the appropriate time during the farming season or by sending messages targeting specific pest outbreaks.

Our results suggest that the demand for extension services is high only when faced with significant crop losses. Considerable rethinking in innovative ways is critical to enhancing the demand for extension services. For instance, improving demand, reminders, and information nudges via farmer schools or workshops with demonstration plots can give the farmers confidence in digital agricultural advice. To break off the Green Revolution era's legacy–the use of more chemical inputs–the extension services must be more proactive in promoting prudent use of harmful inputs and better managing pest pressure.

Though studies using RCTs have poor external validity, the lessons learnt from our intervention in India have broader implications for regions and countries where production risks in agriculture are considerably high. While extension services are already prevalent in many countries, improving their access and timely delivery with new ICT tools can accelerate agricultural development. For instance, sharing photographs electronically (e.g., posters) using Android-based devices to identify and confirm specific agricultural problems would increase the effectiveness of hotline services.

Our analysis's principal limitation is that we compare the hotline impact over two different crop cycles with the shock occurring only in the second crop cycle. We use the difference-in-difference estimation strategy to address the concern, which compares the outcomes before and after the SMD shock and between treatment and control groups. Another limitation of our study is the insufficient statistical power to examine the heterogeneous effect of hotline treatment by caste and land size holdings. More research on improving the extension system to be more inclusive will require a larger sample with an improved research design.

## Conclusion

Traditionally agricultural extension is delivered by extension workers with often no training in science-based agricultural advice, limited budget to manoeuvre farm visits, and poor accountability. With digital agriculture, governments can reorient their policies and budgets to strengthen transmissions of information electronically. The broader coverage of mobile phone networks has enhanced the potential to address the information asymmetries within poor communities at no additional costs. Our paper explores whether we can improve the adoption of best agricultural practices, where information supply and acquisition are less costly, by harnessing the existing communication infrastructure.

Several past studies have examined the impact of access to agricultural information, reporting mixed results and considerable context-dependence. Our paper builds on these studies by going beyond access, suggesting a need to pay close attention to the heterogeneity of crop shocks and information delivery timing. Thus, for crops that faced adverse shocks, the demand for information is likely high, so the information delivery could positively impact productivity. In contrast, for crops with isolated, localized production shocks, the information provided may not affect output due to the low demand for extension services.

## Supporting information

**S1 Fig. Timeline of the intervention.**
(DOCX)

**S1 Table. Selected variable definition in regression analysis.**
(DOCX)

**S2 Table. Differential effect of shocks in the second crop cycle across treatment groups.**
(DOCX)

**S3 Table. Hotline usage.**
(DOCX)

**S4 Table. Control and treatment households–baseline balance check -2013.**
(DOCX)

**S5 Table. Summary of the information received from the KCC helpline.**
(DOCX)

**S6 Table. P-values for the model with controls.**
(DOCX)

**S1 Data. Data and STATA codes used.**
(ZIP)

## Author Contributions

**Conceptualization:** Arjunan Subramanian.

**Data curation:** Arjunan Subramanian.

**Formal analysis:** Arjunan Subramanian.

**Funding acquisition:** Arjunan Subramanian.

**Investigation:** Arjunan Subramanian.

**Methodology:** Arjunan Subramanian.

**Resources:** Arjunan Subramanian.

**Software:** Arjunan Subramanian.

**Supervision:** Arjunan Subramanian.

**Validation:** Arjunan Subramanian.

**Visualization:** Arjunan Subramanian.

**Writing – original draft:** Arjunan Subramanian.

**Writing – review & editing:** Arjunan Subramanian.

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
