## [Decision Letter · Decision Letter 0]

4 Dec 2020

PONE-D-20-33119

Harnessing digital technology to improve agricultural productivity?

PLOS ONE

Dear Dr. Subramanian,

Thank you for submitting your manuscript to PLOS ONE. After careful consideration, we feel that it has merit but does not fully meet PLOS ONE’s publication criteria as it currently stands. Therefore, we invite you to submit a revised version of the manuscript that addresses the points raised during the review process.

The study makes solid contributions to the literature and the authors are encouraged to make the revisions and clarifications outlined by the reviewers. Understanding how to promote and better understand farmer ecologically based practices such as IPM urgently needs studies such as this, so the authors are commended for this work. There does however need to be substantial revisions, notably stats, and clarifications of terms, so I look forward to all of the reviewer comments being addressed. The reviewers have taken considerable time to make detailed comments.

We look forward to receiving your revised manuscript.

Kind regards,

Sieglinde S. Snapp

Academic Editor

PLOS ONE

Journal Requirements:

2. Please include your tables as part of your main manuscript and remove the individual files. Please note that supplementary tables (should remain/ be uploaded) as separate "supporting information" files

Additional Editor Comments (if provided):

There are indeed some issues that need to be addressed as highlighted by the careful reviews. In particular, consider more carefully if this is “investing in new technology” as farmers follow a wide range of practices related to this topic, and some are adopted and dis-adopted over time and space. This needs to be acknowledged and addressed in the introduction and discussion. There are also the issues raised re variety specific, crop-plot - there is some confusion at multiple points as raised by the reviewers, these need to be clarified. Overall, a solid contribution to the literature once the statistical issues raised by the reviewers, and all of these points are fully addressed. Also - please do use the international common name for the crops, eg pigeonpea.

Reviewers' comments:

Reviewer's Responses to Questions

**Comments to the Author**

1. Is the manuscript technically sound, and do the data support the conclusions?

Reviewer #1: Partly

Reviewer #2: Yes

2. Has the statistical analysis been performed appropriately and rigorously? 

Reviewer #1: Yes

Reviewer #2: Yes

3. Have the authors made all data underlying the findings in their manuscript fully available?

Reviewer #1: Yes

Reviewer #2: Yes

4. Is the manuscript presented in an intelligible fashion and written in standard English?

Reviewer #1: No

Reviewer #2: Yes

5. Review Comments to the Author

Reviewer #1: Review of PONE-D-20-33119 “Harnessing digital technology to improve agricultural productivity?”

Overall comments

This paper evaluates the impacts of SMS-based extension advisory services on yield outcomes on a sample of Indian farmers. This is an important question, given the fervor surrounding investments in digital agriculture in developing countries, and the immodest claims for such investments, which still often lack a solid empirical foundation. The experimental set-up has some nice features which address potential bias from self-selection and from treatment spillover. However, the nature of the information treatment and its linkage(s) with outcomes of interest is not clear enough for the analysis to be very compelling. If these aspects of the study can be clarified, I think the paper may make a nice contribution and merit publication in PLOS One.

Major issues

The main shortcoming of the paper is a lack of conceptual clarity about the nature of the information treatments and the outcomes of interest. Please be much clearer about the type of advice that is conjectured to have been most relevant for understanding these results, i.e. what advice did red gram farmers receive when calling in about an incidence of SMD? What are the other types of information provided and how are these different types of information linked to the outcomes of interest in this study? The current discussion around impacts of “better farming information” is far too vague. Somewhat relatedly, the discussion of adoption of “cost-effective” farming practices needs much more clarity for us to understand this. I suggest a section that clearly lays out the types of information provided, and its expected impacts on different outcomes of interest.

Given the importance of timing of advisory services, how is the timing of treatment controlled for in this analysis?

You should provide a fuller description of the intention-to-treat analysis -- i.e. that your analysis uses the initial treatment assignment (not the actual information treatment) as the basis for impact assessment. (At least, if I understand correctly what you did. It is not presented very clearly.) You should clearly show the estimating equation and the terms which constitute the intention-to-treat effect.

Your assertion that the information treatment is “variety specific” needs some unpacking. First, how would the advice for dealing with SMD differ by different red gram varieties? Second, given what we now know about widespread misidentification of varieties by farmers (e.g. Kosmowski et al. 2019, Jaleta et al. 2020), what implications might this have for your analysis?

It is not clear why the information treatment would affect cost of harvesting. Can you explain this? Also, what are interculture costs, and how would these be affected by in-season advisory services?

The review of related literature examining impacts of advisory services is very limited and needs to be expanded. Lots of experimental approaches to impact evaluation of digital advisory services have been coming out recently (e.g. Ayalew et al. 2020, Arouna et al. 2020 – see Norton and Alwang 2020 for a review of others).

What are the limitations of this study? What are the threats to identification in your analysis? What about external validity? These issues should be at least acknowledged.

Finally, if I understand correctly, in the absence of the SMD event, would there have been any detectable effects? Given the crop-specific results, I think not. That result deserves more discussion.

Detailed comments

General advice: use page numbers in submitted manuscripts to facilitate comments!

Abstract: “a high-stakes crop” – tell us what crop this is and what makes it a “high-stakes” crop.

Abstract: “Our findings reveal a [more] nuanced causal relationship than previously suggested, providing an explanation for the existence of mixed results [of prior impact evaluations].” See suggested insertions in brackets.

Abstract: is this a national study? If not, give a few more details of the study scope & sample size.

I think “redgram” should be rendered as “red gram”.

Can you give us an example of the kind of response advice that is given in response to early detection of SMD?

“Careful rouging of the infected plant…” Can you explain what this means? Do you mean roguing?

Many minor grammatical issues (e.g. dropped articles) suggest that professional copyediting may be desirable.

“The usage of the KCC helpline number was high, with 91% of the treated households calling the number to receive a range of information (online Appendix Figure S1)” – this is not clear, as the figure indicates treatment in terms of households, but uptake of advice is given as plot-crops. Is there just one plot-crop per sampled household?

For PLOS ONE’s readership, you should explain how the “intent-to-treat impact” is calculated.

Table 1: what is “Hotline impact”? do you mean use of the hotline, conditional on treatment?

You talk about “investing in new technology” (which I assume is referring to varieties and inputs) but we do not know if these technologies are in fact new for the farmers in this sample.

References

Arouna, Aminou and Michler, Jeffrey D. and Yergo, Wilfried and Saito, Kazuki, One Size Fits All? Experimental Evidence on the Digital Delivery of Personalized Extension Advice in Nigeria (2020). Available at SSRN: https://ssrn.com/abstract=3593878 or http://dx.doi.org/10.2139/ssrn.3593878

Ayalew, H., Jordan, C. and Newman, C., (2020). Site-Specific Agronomic Information and Technology Adoption: A Field Experiment from Ethiopia. Trinity College Dublin, Economics Department. https://EconPapers.repec.org/RePEc:tcd:tcduee:tep0620

Jaleta, M., Tesfaye, K., Kilian, A., Yirga, C., Habte, E., Beyene, H., ... & Erenstein, O. (2020). Misidentification by farmers of the crop varieties they grow: Lessons from DNA fingerprinting of wheat in Ethiopia. Plos one, 15(7), e0235484.

Kosmowski, F., Aragaw, A., Kilian, A., Ambel, A., Ilukor, J., Yigezu, B., & Stevenson, J. (2019). Varietal identification in household surveys: results from three household-based methods against the benchmark of DNA fingerprinting in southern Ethiopia. Experimental Agriculture, 55(3), 371-385.

Norton, G. W., & Alwang, J. (2020). Changes in Agricultural Extension and Implications for Farmer Adoption of New Practices. Applied Economic Perspectives and Policy, 42(1), 8-20.

Reviewer #2: Review of “Harnessing digital technology to improve agricultural productivity?” (PONE-D-20-33119)

1. Is the manuscript technically sound, and do the data support the conclusions?

a. Yes but please see below for some suggestions on how to improve the paper.

b. KCC hotline services section: How do things work when farmers call the KCC hotline? Do they speak live with a Level 1 operator and get answers to their specific questions? And how site-specific is the information provided? This information would greatly improve the reader’s understanding of the nature of the information that is provided by the KCC hotline.

c. Nature of crop pest shock section: It is important to back up the reported yield losses and restorative measures described with adequate citations. There are no citations in this section at present.

d. Minor point but given the international readership of the journal, I suggest using the terms pigeon pea and finger millet throughout (instead of redgram and ragi, respectively).

e. Sample selection section:

i. Why was Gubbi taluk selected and how generalizable are the results of the study to other areas (external validity)? What is the prevalence of mobile phone ownership in the study area?

ii. What specific criteria were used to designate given areas as “predominantly redgram-growing, ragi-growing, and the rest”? How many of the treatment and control villages (and farmers) are in each of these strata?

f. Data section: what exactly is meant by “crop-plots”? More information on the unbalanced panel is also needed in the main text. How many crop-plots are in the baseline survey? Of those, how many are also on the endline survey and how many new crop-plots on the endline survey? What is the breakdown by treatment vs. control and by crop? How many households control these crop-plots?

g. Results section: If redgram was the only focal crop of the study to experience a significant pest shock during the study period, then I don’t follow the logic of this sentence: “The information from the hotline – recommendations based on Integrated Pest Management (IPM) methods – is expected to be equally useful for all crops and against most pests and diseases. Thus, the shock to redgram per se is unlikely to be the cause for differences in the crop yields.” A similar point is made in the Discussion that I disagree with: “However, information on IPM methods from hotlines is equally useful for all crops.” These points also seem to conflict many of the arguments made in the Discussion section in terms of why there are KCC impacts on redgram but not other crops. Had there been a pest shock to the other crops that could be addressed via IPM (or other methods on which technical advice could be provided over the phone), might we expect similar effects?

h. Discussion section: I don’t find the closing paragraph useful or adequately supported with citations. I suggest dropping it and wrapping up the discussion with a concluding sentence or two at the end of the previous paragraph.

2. Has the statistical analysis been performed appropriately and rigorously?

a. Yes, as far as I can tell. (I am not an RCTs expert.)

b. The interpretation of the results throughout the paper as percentage point (pp) effects is incorrect. These are percentage effects, not PP effects, because the dependent variable is in logs. This needs to be corrected throughout the paper wherever results are discussed. See Wooldridge’s Introductory Econometrics textbook (any version) for more info on how to interpret log-level models.

c. Figure 1: it would be more accurate to label the lines treatment and control.

d. It would be helpful to add error bars to Figure 2 and use 2-dimensional (instead of 3-dimensional) columns.

3. Have the authors made all data underlying the findings in their manuscript fully available?

a. Yes, the data, replication code, and instructions are included in a zip file that can be linked to in the supplemental materials.

4. Is the manuscript presented in an intelligible fashion and written in standard English?

a. Yes, the paper is generally well written but there are numerous grammatical errors. The paper (including the abstract) would benefit from careful copy editing.

6. PLOS authors have the option to publish the peer review history of their article (what does this mean?). If published, this will include your full peer review and any attached files.

Reviewer #1: No

Reviewer #2: No

---

## [Author Response · Author response to Decision Letter 0]

15 Feb 2021

Comments in italics. Responses in regular font.

Responses to Editor

• The study makes solid contributions to the literature and the authors are encouraged to make the revisions and clarifications outlined by the reviewers. Understanding how to promote and better understand farmer ecologically based practices such as IPM urgently needs studies such as this, so the authors are commended for this work. There does however need to be substantial revisions, notably stats, and clarifications of terms, so I look forward to all of the reviewer comments being addressed. The reviewers have taken considerable time to make detailed comments.

- We have rewritten the paper addressing all of the points raised by the reviewers. Most notably, we have added two new sections: one, clearly laying out the types of information provided by the hotline and its expected impacts on different outcomes of interest (pages 7-9). Second, illustrate the regression model we estimate and the terms which constitute the intention-to-treat (pages 14 and 15). 

• There are indeed some issues that need to be addressed as highlighted by the careful reviews. In particular, consider more carefully if this is “investing in new technology” as farmers follow a wide range of practices related to this topic, and some are adopted and dis-adopted over time and space. This needs to be acknowledged and addressed in the introduction and discussion.

- As suggested, a section is added in the revised paper on the types of information provided under the sub-title "Technology and management strategies disseminated". We now add the following section to page 7 in the revised paper: "Though SMD is not a new disease, its causal agent has remained elusive to identification and characterization over many decades. Farmers follow a wide range of practices and adopt and dis-adopt technologies over time. Some of the technologies advocated by the hotline were new to a few farmers, while others may have been dis-adopted over time. Thus, we now refer to the hotline information treatment on the inputs use as adoption of best agricultural practices rather than new technologies. 

• There are also the issues raised re variety specific, crop-plot - there is some confusion at multiple points as raised by the reviewers, these need to be clarified. 

- Some pigeon pea variety offers broad-based resistance to SMD, while others are highly susceptible to the disease. For the less tolerant varieties, heavy doses of chemicals were prescribed, while roguing was recommended for resistant varieties. Thus, the information provided by the hotline varied depending on the crop variety. Here crop-plot refers to a parcel of land with single crop demarcated by raised bunds. We incorporate all of these clarifications in the revised paper.

• Also - please do use the international common name for the crops, eg pigeonpea.

- We completely agree. Given the international readership of the journal, we changed the terms pigeon pea and finger millet instead of redgram and ragi.

Comments in italics. Responses in regular font.

Responses to Reviewer #1

Thank you very much for your time and detailed, thoughtful comments, which we believe have significantly improved the paper. Below we present the point-wise response showing how we addressed your comments in the revised paper.

Comments:

• This paper evaluates the impacts of SMS-based extension advisory services on yield outcomes on a sample of Indian farmers. This is an important question, given the fervor surrounding investments in digital agriculture in developing countries, and the immodest claims for such investments, which still often lack a solid empirical foundation. The experimental setup has some nice features which address potential bias from self-selection and from treatment spillover. However, the nature of the information treatment and its linkage(s) with outcomes of interest is not clear enough for the analysis to be very compelling. If these aspects of the study can be clarified, I think the paper may make a nice contribution and merit publication in PLOS One.

- Thank you very much for your constructive comments. We have completely rewritten all parts of the paper. We believe that our significant revisions to the paper clarify the nature of the information treatment and its linkages with outcomes of interest.

Major issues

• The main shortcoming of the paper is a lack of conceptual clarity about the nature of the information treatments and the outcomes of interest. Please be much clearer about the type of advice that is conjectured to have been most relevant for understanding these results, i.e. what advice did red gram farmers receive when calling in about an incidence of SMD? What are the other types of information provided and how are these different types of information linked to the outcomes of interest in this study? The current discussion around impacts of "better farming information" is far too vague. Somewhat relatedly, the discussion of adoption of "cost-effective" farming practices needs much more clarity for us to understand this. I suggest a section that clearly lays out the types of information provided, and its expected impacts on different outcomes of interest.

- We agree. As suggested, a section is added in the revised paper on the types of information provided under the sub-title "Technology and management strategies disseminated". We now add the following section to page 7 in the revised paper: "Though SMD is not a new disease, its causal agent has remained elusive to identification and characterization over many decades. The dynamics of SMD are influenced by many abiotic and biotic factors, diverse agricultural systems, and environmental conditions. Information on off-season survival of virus and mite vector, their spread during the cropping season from crop to crop, and within the crop and variation in disease incidence in a region is critical for understanding disease ecology. It is a challenging task considering that the crop is mainly grown in marginal farming systems with divergent cropping practises. 

Several methods were recommended by the hotline to reduce SMD incidence by using pesticides to delay the onset of infection and disease spread, control through cropping management practices, and host-plant resistance. In the worst cases of the disease, helpline recommended many chemicals to control the mite vector and minimize the disease's spread (online Appendix, Table S5). Correct timing and dosage are critical for effective control of the vector populations. 

The helpline relayed to farmers not to go for continuous cultivation of pigeon pea in the same field and treat the seeds with Trichoderma viride at 4 gram per kg of seed. Farmers were advised to pull out the infected plants and burn them as SMD spreads quickly - roguing or pruning diseased plants were instructed to promote cost-effective treatment. Neem oil spray (1 to 2 ml per litre) or Profenophos (2ml per litre) was recommended instead of expensive, strong chemicals. Farmers were advised not to spray chemicals inadvertently but adopt methods like pruning the infected plants, and only react with chemicals when the pest population reached the optimal threshold. The threshold was described, shown, and discussed with farmers. 

Overall, removing diseased plants from the field (roguing) was the highly recommended and widely adopted SMD management measure. Roguing is most effective, easier to perform, and has a less detrimental effect on productivity when carried out early in crop development. The recommendation is likely to reduce the cost of production relative to the control group. More specifically, it can reduce the cost of insecticides and associated labour use. In contrast, the cost of intercultural operation is likely to rise with the increased labour use for regular inspection and roguing of infected plants. The damage control recommendations can result in greater yield and raise the cost of harvesting relative to the control group." 

• Given the importance of timing of advisory services, how is the timing of treatment controlled for in this analysis?

- A dummy variable T is included in the regression representing the time that takes value 1 for crop cycle 2 in 2014 and 0 for crop cycle 1 in 2013. Please see the section on "Regression model" on page 14 of the revised paper.

• You should provide a fuller description of the intention-to-treat analysis -- i.e. that your analysis uses the initial treatment assignment (not the actual information treatment) as the basis for impact assessment. (At least, if I understand correctly what you did. It is not presented very clearly.) You should clearly show the estimating equation and the terms which constitute the intention-to-treat effect.

- Fully agree. A complete section is now added under "Regression model" on page 14 of the revised paper, showing the estimation equation and the effect of interest (ITT). 

• Your assertion that the information treatment is "variety specific" needs some unpacking. First, how would the advice for dealing with SMD differ by different red gram varieties? 

- Some pigeon pea varieties offer broad-based resistance to SMD, while others are highly susceptible to the disease. For highly sensitive varieties, heavy doses of chemicals were prescribed, whilst roguing was recommended for resistant varieties. Thus, the information provided by the hotline varied depending on the crop variety. Please see page 8 in the revised paper. 

• Second, given what we now know about widespread misidentification of varieties by farmers (e.g. Kosmowski et al. 2019, Jaleta et al. 2020), what implications might this have for your analysis?

- Thanks for bringing our attention to this literature in the African context. However, we found that awareness about crops and crop varieties seemed to be high among Indian farmers. The enumerators recruited to conduct surveys were graduates from the local agricultural university familiar with local farming practices. They were also trained to identify crops, crop varieties, brand names of fertilizers and insecticides, and other farm inputs. We can attribute the knowledge to the extensive network of extension institutions (agricultural universities and research centres), a legacy of the successful Green Revolution era, that brought considerable productivity gains in India. 

• It is not clear why the information treatment would affect cost of harvesting. Can you explain this? Also, what are interculture costs, and how would these be affected by in-season advisory services?

- The information treatment recommendations are likely to raise the cost of intercultural operations with the increased labour use for regular inspection of the crop and roguing of infected plants. The damage control recommendations from the advisory services can result in greater yield and increase the harvesting cost of labour relative to the control group. We include these explanations on page 8 of the revised paper.

• The review of related literature examining impacts of advisory services is very limited and needs to be expanded. Lots of experimental approaches to impact evaluation of digital advisory services have been coming out recently (e.g. Ayalew et al. 2020, Arouna et al. 2020 – see Norton and Alwang 2020 for a review of others).

- We agree with the suggestion of expanding the literature with some recent contributions. Thus, we have added the following in the revised paper (page 3): "Our study contributes to the evolving experimental literature on the impact of digital advisory services in agriculture (12-14). Our results have considerable policy relevance for using information and communication technology (ICT) on development outcomes (15-17). The broader coverage of mobile phone networks can address the information asymmetries within poor communities at no additional costs. Our paper explores whether we can improve the adoption of best agricultural practices by making information supply and acquisition less costly by harnessing the existing communication infrastructure."

• What are the limitations of this study? What are the threats to identification in your analysis? What about external validity? These issues should be at least acknowledged.

- Thank you for drawing our attention to the limitations of the study. The Gubbi sub-district is representative of pigeon pea growing areas across central and southern India. Any generalizations of the results beyond this region may be problematic (page 9). 

- Though studies using RCTs have poor external validity, the lessons learnt from our intervention in India have broader implications for regions and countries where production risks in agriculture are considerably high (page 21). 

- Our analysis's principal limitation is that we compare the hotline impact over two different crop cycles with the shock occurring only in the second crop cycle year. We use the difference-in-difference estimation strategy to address the concern, which compares the outcome before and after the SMD shock and between treatment and control groups. Another limitation of our study is the insufficient statistical power to examine the heterogeneous effect of hotline treatment by caste and land size holdings. More research on improving the extension system to be more inclusive will require a larger sample with improved research design. We have now incorporated the above statements in the revised paper (page 22).

• Finally, if I understand correctly, in the absence of the SMD event, would there have been any detectable effects? Given the crop-specific results, I think not. That result deserves more discussion.

- We agree with you. No detectable effect may be found in the absence of SMD. We find that farmers' access hotline only for the crop that faced a covariate shock and not for crops tackling minor pest problems (See SI Table S3). However, it may be likely that farmers were preoccupied with addressing a more significant issue in pigeon pea, thus ignoring the smaller issues in other crops, despite knowing the effectiveness of hotline information against pest and diseases in any crop. However, a discussion on what happens to the demand for hotline services in the absence of major shock like SMD will require an entirely different experimental setup beyond our paper's scope. 

Detailed comments

• General advice: use page numbers in submitted manuscripts to facilitate comments!

- Thank you for drawing our attention to this; it's an omission which we now correct.

• Abstract: "a high-stakes crop" – tell us what crop this is and what makes it a "high-stakes" crop.

- Pigeon pea is the high-stakes crop because it faced adverse aggregate shock.

• Abstract: "Our findings reveal a [more] nuanced causal relationship than previously suggested, providing an explanation for the existence of mixed results [of prior impact evaluations]." See suggested insertions in brackets.

- Thank you for your suggestion. We have now corrected the Abstract.

• Abstract: is this a national study? If not, give a few more details of the study scope & sample size.

- This is not a national study. Thus, the following statement is added to the Abstract: We conducted our study among 300 farmers in the southern Indian state of Karnataka.

• I think "redgram" should be rendered as "red gram".

- Given the journal's international readership, we changed the terms pigeon pea and finger millet instead of redgram and ragi.

• Can you give us an example of the kind of response advice that is given in response to early detection of SMD?

- Roguing is most effective, easier to perform, and has a less detrimental effect on productivity when carried out early in crop development. The recommendation is likely to reduce the cost of production relative to the control group. More specifically, it will reduce the cost of insecticides and associated labour use (page 8 of the revised paper).

• "Careful rouging of the infected plant…" Can you explain what this means? Do you mean roguing?

- Thank you for drawing our attention to this; it's an omission which we now correct – it is roguing.

• Many minor grammatical issues (e.g. dropped articles) suggest that professional copyediting may be desirable.

- We thoroughly checked and corrected the revised paper for grammatical errors. 

• "The usage of the KCC helpline number was high, with 91% of the treated households calling the number to receive a range of information (online Appendix Figure S1)" – this is not clear, as the figure indicates treatment in terms of households, but uptake of advice is given as plot-crops. Is there just one plot-crop per sampled household?

- These are marginal and small farmers with one or two plots of owned land. Households in our sample typically cultivate one plot-crop per year but some households adopted a different crop in the second year. 

• For PLOS ONE's readership, you should explain how the "intent-to-treat impact" is calculated.

- A complete section is now added in page 14 under "Regression model" showing the estimating equation and the intent-to-treat impact.

• Table 1: what is "Hotline impact"? do you mean use of the hotline, conditional on treatment?

- You are correct – hotline use conditional on treatment. Please see the footnote to Table 1.

• You talk about "investing in new technology" (which I assume is referring to varieties and inputs) but we do not know if these technologies are in fact new for the farmers in this sample.

- Some pigeon pea variety offer broad-based resistance to SMD while others are highly susceptible to the disease. Thus, the information provided by the hotline varied depending on the crop variety. For the resistant type, the information stressed on roguing. For the varieties that are highly susceptible to SMD, the hotline recommended a combination of chemical sprays. The combination of the sprays was based on new scientific evidence. Though SMD is not a new disease, the causal agent is just identified, and disease somewhat understood. Farmers follow a wide range of practices and adopt and dis-adopt technologies over time. Some of the technologies advocated by the hotline were new to a few farmers, while others may have dis-adopted. Thus, we now refer to the hotline information treatment on the inputs use as adoption of best agricultural practices rather than new technologies (page 4). 

• References

Arouna, Aminou and Michler, Jeffrey D. and Yergo, Wilfried and Saito, Kazuki, One Size Fits All? Experimental Evidence on the Digital Delivery of Personalized Extension Advice in Nigeria (2020). Available at SSRN: https://ssrn.com/abstract=3593878 or http://dx.doi.org/10.2139/ssrn.3593878

Ayalew, H., Jordan, C. and Newman, C., (2020). Site-Specific Agronomic Information and Technology Adoption: A Field Experiment from Ethiopia. Trinity College Dublin, Economics Department. https://EconPapers.repec.org/RePEc:tcd:tcduee:tep0620

Jaleta, M., Tesfaye, K., Kilian, A., Yirga, C., Habte, E., Beyene, H., ... & Erenstein, O. (2020). Misidentification by farmers of the crop varieties they grow: Lessons from DNA fingerprinting of wheat in Ethiopia. Plos one, 15(7), e0235484.

Kosmowski, F., Aragaw, A., Kilian, A., Ambel, A., Ilukor, J., Yigezu, B., & Stevenson, J. (2019). Varietal identification in household surveys: results from three household-based methods against the benchmark of DNA fingerprinting in southern Ethiopia. Experimental Agriculture, 55(3), 371-385.

Norton, G. W., & Alwang, J. (2020). Changes in Agricultural Extension and Implications for Farmer Adoption of New Practices. Applied Economic Perspectives and Policy, 42(1), 8-20.

- Thanks for drawing our attention to the recent literature on the topic. In the revised paper, we have incorporated all of the references, including some additional papers. 

Comments in italics. Responses in regular font.

Responses to Reviewer #2

Thank you very much for your time and comments, which we believe have significantly improved the paper. Below we present the point-wise response showing how we addressed your comments in the revised paper.

• KCC hotline services section: How do things work when farmers call the KCC hotline? Do they speak live with a Level 1 operator and get answers to their specific questions? And how site-specific is the information provided? This information would greatly improve the reader's understanding of the nature of the information that is provided by the KCC hotline.

- Under the KCC hotline services, farmers call a typical, toll-free advisory phone number, to access expert advice from Level 1 operators (agricultural graduates) in 13 regional centres. The crop-specific extension information, provided from centres located across the country, is delivered in 21 local languages. Further, queries from Level 1 operators are supported by Level 2 experts, who are in different parts of the country at State Agriculture Universities, Indian Council for Agricultural Research Institutes, and State Department of Agriculture. Farmers speak live with the Level 1 operators and get answers to their questions immediately. If Level 1 operators are unable to provide answers, the calls are passed-on to Level 2 experts who also provide answers promptly. The solutions provided are plot- and crop-specific. We incorporate the above details in page 5 of the revised paper.

• Nature of crop pest shock section: It is important to back up the reported yield losses and restorative measures described with adequate citations. There are no citations in this section at present.

- In the revised paper (page 6), we have added four new references to back up the reported yield losses and restorative measures:

 22. L. Kumar, T. Jones, D. Reddy, A novel mite-transmitted virus with a divided RNA genome closely associated with pigeonpea sterility mosaic disease. Phytopathology 93 (2), 71–81 (2003).

 23. K. Ganapathy, M. Gowda, B. Ajay, S. Venkatesha, B. Gnanesh, S. Gomashe, P. Babu, G. Girish, P. Prasad, G. Veerakumar, J. Patil, Inheritance studies of sterility mosaic disease (SMD) resistance in vegetable type pigeon pea (Cajanus cajan (L.) Millsp.). Australian Journal of Crop Science 63, 1154–58 (2012).

 24. L. Kumar, T. Jones, F. Waliyar, Biology, etiology and managment of pigeon pea sterility mosaic disease. Annual Review of Plant Pathology 3, 1-24 (2004).

 25. T. Jones, L. Kumar, K. Saxena, N. Kulkarni, V. Muniyappa, F. Waliyar, Sterility Mosaic Disease - the 'Green Plague' of pigeon pea: advances in understanding the etiology, transmission and control of a major virus disease. Plant Disease, 88 (2), 436–445 (2004).

• Minor point but given the international readership of the journal, I suggest using the terms pigeon pea and finger millet throughout (instead of redgram and ragi, respectively).

- We agree with you. Given the journal's international readership, we changed the terms pigeon pea and finger millet instead of redgram and ragi.

• Sample selection section: Why was Gubbi taluk selected and how generalizable are the results of the study to other areas (external validity)? What is the prevalence of mobile phone ownership in the study area?

- While Tumkur was one of the districts preselected for implementing a large-scale agricultural program, we chose Gubbi for two reasons. (a) After completing the baseline survey under the agricultural program, we noticed the early signs of an SMD outbreak in Gubbi. (b) Access to the KCC hotline is available, but farmers are unaware of its services. The Gubbi sub-district is representative of pigeon pea growing areas across central and southern India. Any generalizations of the results beyond this region may be problematic. A study from the region suggests that only 87% of the households owned mobile phones. However, our baseline survey indicates 100% ownership among sample households owning one or more mobile phones (SI Table S4). We incorporate the above details in page 9 of the revised paper.

• What specific criteria were used to designate given areas as "predominantly redgram-growing, ragi-growing, and the rest"? How many of the treatment and control villages (and farmers) are in each of these strata?

- Two crops – pigeon pea and finger millet – proved significant in terms of area and main staples. To guarantee the desired heterogeneity in terms of crops, we stratified villages by their share of each crop area to total cultivated into three predominantly pigeon pea-growing, finger millet-growing, and the rest. Twenty-eight villages, primarily growing pigeon pea, were randomly allocated to treatment with twenty-five to control. We randomly assigned twenty villages with a significant share of finger millet to treatment and twenty-two villages to control. Rest of the twenty-six villages are split with eleven randomly allocated to treatment and the rest to control. The households in each stratum consist of 90 pigeon pea farmers, 107 finger millet farmers, and 38 growing other crops (see page 10 in the revised paper).

• Data section: what exactly is meant by "crop-plots"? More information on the unbalanced panel is also needed in the main text. How many crop-plots are in the baseline survey? Of those, how many are also on the endline survey and how many new crop-plots on the endline survey? What is the breakdown by treatment vs. control and by crop? How many households control these crop-plots?

- Here crop-plot refers to a parcel of land with a single crop demarcated by raised bunds.

- In the baseline survey, there are 193 crop-plots which also appears in the endline survey along with 42 additional crop-plots. Figure 1 in the revised paper shows that 160 treatment farmers cultivate 297 crop-plots, and 75 control farmers cultivate 131 crop-plots. The breakdown of crop-plots by type of crops are 148 pigeon pea, 208 finger millet, and 72 other crops. We include these details in the revised paper on page 11. 

• Results section: If redgram was the only focal crop of the study to experience a significant pest shock during the study period, then I don't follow the logic of this sentence: "The information from the hotline – recommendations based on Integrated Pest Management (IPM) methods – is expected to be equally useful for all crops and against most pests and diseases. Thus, the shock to redgram per se is unlikely to be the cause for differences in the crop yields." A similar point is made in the Discussion that I disagree with: "However, information on IPM methods from hotlines is equally useful for all crops." These points also seem to conflict many of the arguments made in the Discussion section in terms of why there are KCC impacts on redgram but not other crops. Had there been a pest shock to the other crops that could be addressed via IPM (or other methods on which technical advice could be provided over the phone), might we expect similar effects?

- You make a very interesting point. Indeed, pigeon pea was the only focal crop to experience a significant pest shock. Non-pigeon pea crops had a regular agricultural season with many minor pest and disease issues such as mites, fungal infections, etc. The IPM methods over the hotline can be equally effective against these minor pests and diseases, but farmers chose to access hotline primarily to address SMD in pigeon pea (See SI Table S3). The low demand for hotline by farmers who cultivated only non-pigeon pea suggests that farmers value the extension information more when faced with the risk of crop loss. No detectable effect may be found in the absence of SMD. We find that farmers' access hotline only for the crop that faced a covariate shock and not for crops tackling minor pest problems. However, it may be likely that farmers were preoccupied with addressing a more significant issue in pigeon pea, thus ignoring the smaller issues in other crops, despite knowing the effectiveness of hotline information against pest and diseases in any crop. However, a discussion on what happens to the demand for hotline services in the presence of major shock to non-pigeon pea crops will require an entirely different experimental setup beyond our paper's scope. 

• Discussion section: I don't find the closing paragraph useful or adequately supported with citations. I suggest dropping it and wrapping up the discussion with a concluding sentence or two at the end of the previous paragraph.

- As suggested, we dropped the closing paragraph and introduced a paragraph discussing the study's limitations, as suggested by the first reviewer. The closing paragraph is as follows: Our analysis's principal limitation is that we compare the hotline impact over two different crop cycles with the shock occurring only in the second crop cycle year. We use the difference-in-difference estimation strategy to address the concern, which compares the outcome before and after the SMD shock and between treatment and control groups. Another limitation of our study is the insufficient statistical power to examine the heterogeneous effect of hotline treatment by caste and land size holdings. More research on improving the extension system to be more inclusive will require a larger sample with improved research design. Please see pages 15 and 21 for details.

• The interpretation of the results throughout the paper as percentage point (pp) effects is incorrect. These are percentage effects, not PP effects, because the dependent variable is in logs. This needs to be corrected throughout the paper wherever results are discussed. See Wooldridge's Introductory Econometrics textbook (any version) for more info on how to interpret log-level models.

- Thank you for drawing our attention to this; it's an omission which we now correct throughout the document.

• Figure 1: it would be more accurate to label the lines treatment and control.

- Thanks for the suggestion. We now label the lines as treatment and control in Figure 1.

• It would be helpful to add error bars to Figure 2 and use 2-dimensional (instead of 3-dimensional) columns.

- Thank you for the suggestion. The error bars are added, and the figure is changed to 2-dimension.

• Yes, the paper is generally well written but there are numerous grammatical errors. The paper (including the abstract) would benefit from careful copy editing.

- We thoroughly checked and corrected the paper for grammatical errors.

---

## [Decision Letter · Decision Letter 1]

30 Mar 2021

PONE-D-20-33119R1

Harnessing digital technology to improve agricultural productivity?

PLOS ONE

Dear Dr. Subramanian,

Thank you for submitting your manuscript to PLOS ONE. After careful consideration, we feel that it has merit but does not fully meet PLOS ONE’s publication criteria as it currently stands. Therefore, we invite you to submit a revised version of the manuscript that addresses the points raised during the review process.

Both reviewers are substantially pleased with the revised manuscript. You have addressed many of the concerns. However, there are some points which still need to be addressed, and please pay close attention especially to reviewer #1 substantive points such as the need to provide more context, citing more literature, and close reading of the manuscript as it still has grammar errors and is unclear in places.

An important point is how to access the data and Stata code, these must be available in order to publish and the reader is not clearly pointed to where they can access these.

We look forward to receiving your revised manuscript.

Kind regards,

Sieglinde S. Snapp

Academic Editor

PLOS ONE

Journal Requirements:

Reviewers' comments:

Reviewer's Responses to Questions

**Comments to the Author**

1. If the authors have adequately addressed your comments raised in a previous round of review and you feel that this manuscript is now acceptable for publication, you may indicate that here to bypass the “Comments to the Author” section, enter your conflict of interest statement in the “Confidential to Editor” section, and submit your "Accept" recommendation.

Reviewer #1: (No Response)

Reviewer #2: All comments have been addressed

2. Is the manuscript technically sound, and do the data support the conclusions?

Reviewer #1: Yes

Reviewer #2: (No Response)

3. Has the statistical analysis been performed appropriately and rigorously? 

Reviewer #1: Yes

Reviewer #2: (No Response)

4. Have the authors made all data underlying the findings in their manuscript fully available?

Reviewer #1: No

Reviewer #2: (No Response)

5. Is the manuscript presented in an intelligible fashion and written in standard English?

Reviewer #1: Yes

Reviewer #2: (No Response)

6. Review Comments to the Author

Reviewer #1: PONE-D-20-33119-R1 “Harnessing digital technology to improve agricultural productivity?”

Overall comments

This revision is improved over the previous version. The methods in particular are much better articulated and described. However, there are still some issues to be addressed.

In particular, you still need to better situate this paper within the literature. You say on page 3 “Our study contributes to the evolving experimental literature on the impact of digital advisory services in agriculture (12-14).” But you do not follow up by telling us what this contribution is, relative to the existing literature. Is it simply the same research question and methods applied to a new empirical context (i.e. a different intervention in a different geographical setting)? Or something different? In general, you should try to help readers know how to cite your paper. Is it simply another example of evidence for a positive impact of advisory services? Or something a bit more nuanced? Relatedly, in the discussion or conclusions, you may compare/contrast your main analytical conclusions with those of other empirical papers in the literature.

Furthermore, the main analytical conclusions are not always clear. For example, in the discussion on page 21, you state “This study documents that ICT tools do not unequivocally increase productivity, so merely scaling up the ICT infrastructure for improving access to extension information may be insufficient to enhance agricultural productivity.” But you do not tell us what is required in addition to (or in place of) scaling up. As a result, it is hard to know what to make of this conclusion. Similarly, on page 21, you state: “While extension services are already prevalent in many countries, improving their access and timely delivery with new ICT tools can accelerate agricultural development.” Are you simply advocating for a move from analog to digital extension methods? Or something more specific? Basically, as currently written, it is not clear what conclusions the reader should come away with.

Finally, I could not see how or where to access the data and Stata code, although the submitted materials indicate that these are available as supplementary materials.

Detailed comments

The paper still needs proofing! Some examples from the abstract (“we randomly distributed hotline number” missing an article; “results show that eliminating informational inefficiencies increase...” where the last word should be “increases”).

P3: “Our results have considerable policy relevance…” If you claim to have policy relevance, you must say what this is very explicitly. What specific new insight or insights for policy have been gained from this study?

P3: “The broader coverage of mobile phone networks can address the information asymmetries within poor communities at no additional costs.” What does this mean? Are you suggesting that simply adding more cell phone towers will reduce information asymmetries facing farmers? That does not make sense.

P21: “Considerable rethinking in innovative ways is critical to enhancing the demand for extension services.” What does this mean?

Reviewer #2: (No Response)

7. PLOS authors have the option to publish the peer review history of their article (what does this mean?). If published, this will include your full peer review and any attached files.

Reviewer #1: No

Reviewer #2: No

---

## [Author Response · Author response to Decision Letter 1]

11 May 2021

Comments in italics. Responses in regular font.

Responses to Reviewer #1

Thank you very much for your time, and further comments, which we believe helped polish our paper's contribution. Below we present the point-wise response showing how we addressed your comments in the revised paper.

Comments:

Overall comments

• This revision is improved over the previous version. The methods in particular are much better articulated and described. However, there are still some issues to be addressed.

• In particular, you still need to better situate this paper within the literature. You say on page 3 "Our study contributes to the evolving experimental literature on the impact of digital advisory services in agriculture (12-14)." But you do not follow up by telling us what this contribution is, relative to the existing literature. Is it simply the same research question and methods applied to a new empirical context (i.e. a different intervention in a different geographical setting)? Or something different? In general, you should try to help readers know how to cite your paper. Is it simply another example of evidence for a positive impact of advisory services? Or something a bit more nuanced? 

- P3-5: In the Introduction, we summarized our main results and then listed two contributions of our paper. The following are the additions to the revised paper: 

- Our main result shows that access to better farming information on damage control increases crop yield by 31% with the adoption of cost-effective and improved farming practices. Additionally, we have two key findings from the disaggregate analysis. First, access to the hotline increases pigeon pea yield by 87% for the treatment farmers relative to the control group with no access to the hotline. Second, though the overall costs of cultivation increased by 39% from intervention, profits for the treatment farmers cultivating pigeon pea are higher by 70%. The observed impact is understandable given the nature of pest shock in the pigeon pea crop and the timing and significance of the expert advice received via the hotline in reducing the losses for the treatment group.

- Our study contributes to the evolving literature on the impact of digital advisory services in agriculture in two ways. First, we present new evidence on improving the delivery of agricultural extension services in developing countries (12-14). In existing experimental studies, the provision of agricultural advisory services is still heavily supply-driven (13). However, observational studies have shown that the demand for extension services may not be high (15, 16). For instance, a recent study on Malawi suggests that access to agricultural advice does not necessarily lead to greater crop productivity (17). Those quality extension services, measured in terms of the farmer's perceived usefulness and relevance of the advice, are a significant predictor of agricultural productivity at the plot level. Thus, with low demand, the supply-side intervention with even good quality extension services is unlikely to influence the outcome, though also dependent on the type of crop and nature of crop shock. Our intervention provides access to reliable extension information and differentiates the demand by crop shock to gauge the impact on agricultural productivity. Our paper offers the first credible evidence showing that the need for extension services is high only when faced with significant crop losses.

- Second, we provide evidence that failure to account for heterogeneity in crop-specific shocks may be a limiting factor in adopting improved technology. This result suggests that customized shock-specific recommendations could improve crop productivity relative to existing generic recommendations. There is no empirical study to the best of our knowledge that assesses whether information accounting for heterogeneity in crop-specific shocks results in higher yields and profits. While extension services are already prevalent in many countries, improving their access and timely delivery with customized information to overcome idiosyncratic crop shocks can accelerate agricultural productivity.

• Relatedly, in the discussion or conclusions, you may compare/contrast your main analytical conclusions with those of other empirical papers in the literature.

- P23-24: While listing the contributions in the Introduction as stated above, we also try to compare/contrast our main results to the existing studies. In addition, we add a "Conclusion" section as suggested. 

Conclusion

 Traditionally agricultural extension is delivered by extension workers with often no training in science-based agricultural advice, limited budget to manoeuvre farm visits and poor accountability. With digital agriculture, governments can reorient their policies and budgets to strengthen transmissions of information electronically. The broader coverage of mobile phone networks has enhanced the potential to address the information asymmetries within poor communities at no additional costs. Our paper explores whether we can improve the adoption of best agricultural practices by making information supply and acquisition less costly by harnessing the existing communication infrastructure.

 Several past studies have examined the impact of access to agricultural information, reporting mixed results and considerable context-dependence. Our paper builds on these studies by going beyond access, suggesting a need to pay close attention to the heterogeneity of crop shocks and information delivery timing. Thus, for crops that faced adverse shocks, the demand for information is likely high, so the information delivery could positively impact productivity. In contrast, for crops with moderate shock, provisioning of information may not affect output due to the low demand for extension services. Our results reveal a more nuanced causal relationship than previously suggested, explaining the existence of mixed results.

• Furthermore, the main analytical conclusions are not always clear. For example, in the discussion on page 21, you state "This study documents that ICT tools do not unequivocally increase productivity, so merely scaling up the ICT infrastructure for improving access to extension information may be insufficient to enhance agricultural productivity." But you do not tell us what is required in addition to (or in place of) scaling up. As a result, it is hard to know what to make of this conclusion.

- P22: We clarified as 'Locally customized information delivered at the appropriate time during the farming season or sending messages targeting specific pest outbreaks can augment the demand for extension services.' 

• Similarly, on page 21, you state: "While extension services are already prevalent in many countries, improving their access and timely delivery with new ICT tools can accelerate agricultural development." Are you simply advocating for a move from analog to digital extension methods? Or something more specific? Basically, as currently written, it is not clear what conclusions the reader should come away with.

- P23: To clarify, we added the following statement in the revised paper: 'For instance, sharing photographs electronically (e.g., posters) using Android-based devices to identify and confirm specific agricultural problems would increase the effectiveness of hotline services.' 

• Finally, I could not see how or where to access the data and Stata code, although the submitted materials indicate that these are available as supplementary materials.

- We deposited all the dataset and the STATA code required to replicate the results on the PLOS ONE website as part of the submission of the manuscript. It is available on their website, but we assume that it will be linked to the Supplementary materials once the paper is published. 

Detailed comments

• The paper still needs proofing! Some examples from the abstract ("we randomly distributed hotline number" missing an article; "results show that eliminating informational inefficiencies increase..." where the last word should be "increases").

- Thanks for the examples. The other corrections completed in the revised paper are as follows:

- P2: ".. to control spillover effects and contaminate the control group."

- P3: "More than just access to information, the timing of the delivery of expert advice may be crucial in farming.". Here "may be" is moved from the beginning of the sentence to towards the end.

- P8: "However, on average, farmers have 30 years…".

- P9: "..the hotline…". Here "the" was added. Missing an article was a common mistake throughout the paper that is now corrected.

Here are few examples. There are too many corrections made in the revised paper for it to be all listed here.

• P3: "Our results have considerable policy relevance…" If you claim to have policy relevance, you must say what this is very explicitly. What specific new insight or insights for policy have been gained from this study?

- P23-24: Instead of saying "policy relevance," which we deleted from the revised paper, we have now explicitly stated the relevance. The following text has been added: "Traditionally agricultural extension is delivered by extension workers with often no training in science-based agricultural advice, limited budget to manoeuvre farm visits and poor accountability. With digital agriculture, governments can reorient their policies and budgets to strengthen transmissions of information electronically."

• P3: "The broader coverage of mobile phone networks can address the information asymmetries within poor communities at no additional costs." What does this mean? Are you suggesting that simply adding more cell phone towers will reduce information asymmetries facing farmers? That does not make sense.

- P24: We are not suggesting but pointing out the following in the revised paper: "The broader coverage of mobile phone networks has enhanced the potential to address the information asymmetries within poor communities at no additional costs." Please note that in the revised paper, we have moved this sentence to page 24.

• P21: "Considerable rethinking in innovative ways is critical to enhancing the demand for extension services." What does this mean?

- P22-23: We clarified by elaborating on how the demand for extension services can be enhanced in the revised paper. We incorporated the following text in the revised paper: "To enhance demand, reminders, and information nudges via farmer schools or workshops with demonstration plots can give the farmers confidence in digital agricultural advice.".

---

## [Decision Letter · Decision Letter 2]

25 May 2021

PONE-D-20-33119R2

Harnessing digital technology to improve agricultural productivity?

PLOS ONE

Dear Dr. Subramanian,

Thank you for these revisions. While one of the referees is happy, the other referee points out a few remaining areas where more clarity is needed. The referee also gives very valuable advise on how to make the the key contribution stand out more, which should be rewarded with more citations, so I think it would be great if you could address these few remaining comments. If you do so, I will look at these myself and if I deem they are sufficiently address, I can accept for publication.

We look forward to receiving your revised manuscript.

Kind regards,

Bjorn Van Campenhout, Ph.D.

Academic Editor

PLOS ONE

Journal Requirements:

Reviewers' comments:

Reviewer's Responses to Questions

**Comments to the Author**

1. If the authors have adequately addressed your comments raised in a previous round of review and you feel that this manuscript is now acceptable for publication, you may indicate that here to bypass the “Comments to the Author” section, enter your conflict of interest statement in the “Confidential to Editor” section, and submit your "Accept" recommendation.

Reviewer #1: (No Response)

Reviewer #2: All comments have been addressed

2. Is the manuscript technically sound, and do the data support the conclusions?

Reviewer #1: Yes

Reviewer #2: (No Response)

3. Has the statistical analysis been performed appropriately and rigorously? 

Reviewer #1: Yes

Reviewer #2: (No Response)

4. Have the authors made all data underlying the findings in their manuscript fully available?

Reviewer #1: Yes

Reviewer #2: (No Response)

5. Is the manuscript presented in an intelligible fashion and written in standard English?

Reviewer #1: Yes

Reviewer #2: (No Response)

6. Review Comments to the Author

Reviewer #1: I have provided comments in an attached PDF document. I think the paper is technically sound, although the nature of the contribution remains poorly articulated.

Reviewer #2: (No Response)

7. PLOS authors have the option to publish the peer review history of their article (what does this mean?). If published, this will include your full peer review and any attached files.

Reviewer #1: No

Reviewer #2: No

---

## [Author Response · Author response to Decision Letter 2]

31 May 2021

Comments in regular font. Responses in red.

Responses to Reviewer #1

Thank you very much for your time and further comments, which we believe helped polish our paper's contribution. Below we present the point-wise response showing how we addressed your comments in the revised paper.

Comments:

 I appreciate that the author tried to respond in good faith to the previous requests. However, it seems that the author rushed these revisions, as they often entail incomplete sentences (e.g., p 23) or very vague assertions. The manuscript text (including the revised sections) remains imprecise in many places, which makes for frustrating reading. Examples from the new paragraphs which were added in response to the request to clarify the contribution (p 4-5): 

"Those quality extension services, measured in terms of the farmer's perceived usefulness and relevance of the advice, are a significant predictor of agricultural productivity at the plot level." What are "those quality extension services"? Not at all clear what this means. (I think you want to say something like: Farmer perceptions of the usefulness and relevance of received advice are significant positive correlates of agricultural productivity outcomes.) 

- In the revised paper, we further clarified accepting changes suggested by you. The changes are as follows: "Farmer perceptions of the usefulness and relevance of received advice are significant positive correlates of agricultural productivity outcomes.".

"Thus, with low demand, the supply-side intervention with even good quality extension services is unlikely to influence the outcome, though also dependent on the type of crop and nature of crop shock." What outcome? Adoption of advice? Yield? Something else?

- Here, the outcome is the crop yield. Thus, we changed the sentence to as follows: "Thus, with low demand, the supply-side intervention with even good quality extension services is unlikely to influence crop yield, though also dependent on the type of crop and nature of crop shock." 

"Our intervention provides access to reliable extension information and differentiates the demand by crop shock to gauge the impact on agricultural productivity." There are so many things going on in this sentence that its hard to parse. What does "reliable" mean in this context? What does it mean to differentiate demand by crop shock?

- Indeed, we agree; thus, this sentence is replaced with the following: "Our intervention provides advisory recommendations, delivered via a hotline, which is tailored to time- and crop-specific shocks to gauge the impact on agricultural productivity."

"Our paper offers the first credible evidence showing that the need for extension services is high only when faced with significant crop losses." I think you are confusing "need" with "demand". Furthermore, the claim itself is so broad that it's hard to take seriously. All farmers face crop losses. A much clearer and more defensible claim would be to say something like: Our results show that the demand for extension services increases when extension messaging directly relates to specific shocks, and is provided during the window of time in which responses to those shocks are feasible.

- We accepted your suggestion and replaced it with, "Our results show that the demand for extension services increases when extension messaging relates directly to specific shocks and is provided during the window of time in which responses to those shocks are feasible."

I think your whole point should be much simpler: your analysis indicates that advisory recommendations, delivered via a hotline, which are tailored to time- and crop-specific shocks are associated with greater impacts than generic recommendations. This is important and sufficient. It just needs to be stated clearly. You might also note that your results are consistent in spirit with other evaluations of tailored recommendations already cited (which indicate that tailoring recommendations to geographic and/or farmer-specific needs are associated with greater impacts than generic recommendations). However, you should more clearly acknowledge that your study focuses on a hotline service, and differs from many other evaluations of ICT-enabled advisory services which focus on digital advisory services provided via extension agents in the field.

- Thank you for the suggestion. We rewrite the entire section on page 5. The following are the changes: "Second, we provide evidence that failure to account for heterogeneity in crop-specific shocks may be a limiting factor in adopting improved technology. There is no empirical study to the best of our knowledge that assesses whether information accounting for heterogeneity in crop-specific shocks result in higher yields and profits. Our result suggests that customized shock-specific recommendations could improve crop productivity relative to existing generic recommendations (18, 19). Our result is consistent with the recent evaluation suggesting that personalized, localized advice is far more effective than blanket recommendations (13,14). Our study differs from these studies that focus on digital advisory services provided via extension agents in the field, while our evaluation utilizes hotline phone services." 

Detailed comments: 

P 23: This new sentence is incomplete: "Locally customized information delivered at the appropriate time during the farming season or sending messages targeting specific pest outbreaks."

- We rewrite this sentence as follows in the revised paper: "For augmenting the demand for extension services, hotlines can deliver locally customized information at the appropriate time during the farming season or by sending messages targeting specific pest outbreaks."

P 25: "In contrast, for crops with idiosyncratic shocks, the information provided may not affect output due to the low demand for extension services." What do you mean by idiosyncratic shocks? Do you mean shocks that are so localized that it is not possible to program advice for? Or are you basically referring simply to normal production conditions? 

- Here, we mean localized production shocks. Thus, we replaced "idiosyncratic" with "isolated, localized production".

P 15: "Our results reveal a more nuanced causal relationship than previously suggested, explaining the existence of mixed results." I do not think this is true. What your study does is focus on a hotline which provides advice on how to respond to disease, which is categorically different from advisory services which focus on nutrient management or other routine aspects of crop husbandry. To say that your study is more nuanced than other studies in the literature is to mischaracterize the salient differences between your study and prior studies.

- We agree with your suggestion, thus, removed this statement from the text.

The abstract needs work to be coherent and understandable. For example: 

"Recent evidence suggests both positive and insignificant ways in which SMS-based agricultural information could affect farming outcomes." → Do you mean both positive and negative, or both significant and insignificant?

- We mean significant and insignificant ways. Thus, the text is changed in the abstract (page 1) and introduction (page 2).

 "Using methods from experimental economics, we randomly distributed the hotline number to generate exogenous variation in the access to farming information." → You need to describe "the hotline number" before this sentence for us to understand it. 

- We introduced the following sentence that describes the hotline services. "Hotline services provide rapid, unambiguous information by agricultural experts over the phone, which are tailored to time- and crop-specific shocks." 

 "Our findings reveal a more nuanced causal relationship than previously suggested, explaining the existence of mixed results." → I don't think you can claim to explain mixed results of impact evaluations of digital extension tools in a literature that spans many different types of tools and contexts. This may be part of an explanation, but you cannot claim to categorically explain the mixed empirical results in the literature without further qualifications. 

- We agree with your suggestion thus replaced the sentence with the following: "Our findings reveal that advisory recommendations customized to time- and crop-specific shocks are associated with greater impact on agricultural productivity."

---

## [Editor Report · Decision Letter 3]

4 Jun 2021

Harnessing digital technology to improve agricultural productivity?

PONE-D-20-33119R3

Dear Arjunan,

Thank you for taking the time address the last remaining concerns of the reviewer. I can now accept this article. Congratulations and thank you for supporting open science!

Kind regards,

Bjorn Van Campenhout, Ph.D.

Academic Editor

PLOS ONE

---

## [Editor Report · Acceptance letter]

16 Jun 2021

PONE-D-20-33119R3 

Harnessing digital technology to improve agricultural productivity? 

Dear Dr. Subramanian:

I'm pleased to inform you that your manuscript has been deemed suitable for publication in PLOS ONE. Congratulations! Your manuscript is now with our production department. 

Kind regards, 

on behalf of

Dr. Bjorn Van Campenhout 

Academic Editor

PLOS ONE